# Archetypal Analysis and DEA Model, Their Application on Financial Data and Visualization with PHATE

**DOI:** 10.3390/e24010088

**Published:** 2022-01-05

**Authors:** Urszula Grzybowska, Marek Karwański

**Affiliations:** Department of Applied Informatics and Mathematics, Warsaw University of Life Sciences, SGGW, 02-787 Warsaw, Poland; urszula_grzybowska@sggw.edu.pl

**Keywords:** archetypal analysis, DEA, PHATE, data visualization

## Abstract

One of the goals of macroeconomic analysis is to rank and segment enterprises described by many financial indicators. The segmentation can be used for investment strategies or risk evaluation. The aim of this research was to distinguish groups of similar objects and visualize the results in a low dimensional space. In order to obtain clusters of similar objects, the authors applied a DEA BCC model and archetypal analysis for a set of companies described by financial indicators and listed on the Warsaw Stock Exchange. The authors showed that both methods give consistent results. To get a better insight into the data structure as well as a visualization of the similarities between objects, the authors used a new approach called the PHATE algorithm. It allowed the results of DEA and archetypal analysis to be visualized in a low dimensional space.

## 1. Introduction

One of the goals of macroeconomic analysis is to rank and segment enterprises described by numerous indicators in terms of their economic efficiency. There are many classical methods that can be used to do this, such as, e.g., clustering methods.

One of the techniques developed for ranking and grouping of economic entities is data envelopment analysis (DEA) proposed first in 1978 and described in [1]. The values obtained in DEA models can be used to organize companies in terms of their effectiveness, then segmentation can be performed, i.e., companies can be divided into relatively homogeneous groups.

There is another approach that can be used for the description of firms—archetypal analysis, introduced in 1994 by Breiman and Cutler and described in [2] and newly applied in [3,4,5,6]. While segmentation is the grouping of similar objects, the analysis of archetypes focuses on the identification of companies (archetypes) spanning the indicator space, i.e., the space between extreme objects treated as a frame. Then other objects are treated as weighted averages of the archetypes. These archetypal companies can be understood as the so-called trend makers and thus used for economic forecasting and analysis.

Adequate visualization is very helpful in the analysis of multidimensional objects. The main task is to choose an appropriate representation of features. Common representations based on dimensionality reduction, such as multivariate scaling or principal factors, as well as simplex graphs with vertices at extreme points, often lead to images that are difficult to interpret. We propose application of PHATE for data presentation in a low dimension space. PHATE [7] is a relatively new approach of constructing a space with a smaller number of dimensions which has proved to give good results in many fields for visualizing the results but the application of PHATE in economics is not known to us.

The aim of the study was to combine the results of analyses with the use of DEA and archetype models. The presentation of the results of both models with help of PHATE makes it possible to find similarities of objects and give background for further studies.

The paper is organized as follows. In Section 2, we describe applied methods and highlight our research design. In Section 3, the data are presented. In Section 4, we describe the results of our analysis. In Section 5, we present our conclusions and provide comments.

## 2. Methods

### 2.1. Data Envelopment Analysis

DEA is a mathematical programming tool for evaluating the performance of a set of objects called decision making units (DMUs). It can be applied to evaluating the performance of objects in case the variables that describe them can be divided into inputs and outputs. The method calculates the performance rating, the $\theta_{o}$ score for each DMU and the performance benchmark set (a peer group of efficient objects), which allows for the identification of the so-called inefficient DMUs. In many DEA models, the DMUs with the efficiency score $\theta_{o}$ equal to 1 are called efficient. Calculation of efficiency can be helpful in improving productivity and performance of an inefficient DMU.

In our calculations we have applied input-oriented BCC model [8]. The model is formulated separately for each DMUo, o=1, 2,…n, of n DMUs, $x_{ij},\;i=1,\;2,\ldots m$ inputs and $y_{rj},\;r=1,\;2,\ldots s$ outputs, in the following way:$$\theta_{0}^{*}=min\theta_{0}\;\mathrm{subject}\;\mathrm{to}$$
(1)$$\sum_{j=1}^{n}x_{ij}\lambda_{jo}\leq \theta_{o}x_{io}\;i=1,2,\ldots ,m$$
(2)$$\sum_{j=1}^{n}y_{rj}\lambda_{jo}\geq y_{ro}\;r=1,2,\ldots ,s\;$$
(3)$$\sum_{j=1}^{n}\lambda_{jo}=1,\lambda_{jo}\geq 0\;j=1,2,\ldots ,n$$
$\lambda_{jo}$ denotes effectivity weights. For the purpose of the research, we have concentrated efforts not on efficiency measure but on distinguishing groups of homogeneous DMUs [9]. DEA is a very popular tool in economic research and numerous papers are published annually about its applications.

### 2.2. Archetypal Analysis

The aim of archetypal analysis is to find representations of objects in a multidimensional data set that would provide reference for other observations. Archetypes are some extreme, not necessarily existing, observations. Each object can be represented as a convex combination of archetypes. Archetypal analysis as a statistical tool to describe multidimensional observations was introduced in 1994 by Breiman and Cutler [2]. It enables both visualization of objects in a low dimensional space and giving reference for other objects. It has found widespread applications in recent years. Archetypes are derived as solutions to an iterative nonlinear optimization algorithm that minimizes RSS (residual sum of squares), i.e., average distance between observations and archetypes [2,3,4].

In the problem formulation we denote by X an n×m matrix of n observations described by m attributes. The aim is to find a k×m matrix Z defining k archetypes. More precisely we are looking for an n×k matrix Z and coefficients that minimize the RSS:(4)$$RSS=∥X-\alpha Z^{T}∥{}_{2}\;\;\;\;,where\;Z=X^{T}\beta \;$$
the optimized coefficients $\alpha_{ij}$ are weights (coefficients) of linear combinations of archetypes for each observation, $\sum_{j=1\;}^{k}\alpha_{ij}=1,\;\alpha_{ij}\geq 0$, i=1,…,n, the optimized coefficients $\beta_{ij},\;\sum_{i=1}^{n}\beta_{ij}=1\;,\;\beta_{ij}\geq 0\;$, are weights (coefficients) of archetypes in the space of observations and $∥\;{∥}_{2}$ is the spectral norm of a matrix.

Equation (4) approximates data with convex combinations of archetypes, $X=\alpha Z^{T}$ and archetypes are convex combinations of data points, $Z=X^{T}\beta$. The optimization procedure concerns also the hyperparameter k, the number of archetypes. Thanks to the development of computational tools, archetypal analysis has gained much attention in recent years [4,5,6].

### 2.3. PHATE

PHATE (potential of heat-diffusion for affinity-based trajectory embedding) is a novel, affinity-preserving embedding of a multidimensional data set into a two or three- dimensional space that preserves local and global properties of the data structure first proposed in 2017 by Moon et al. [7]. It was designed as an answer to an increasing need to visualize, explore and interpret high dimensional biological data, mainly genetical data. The key idea is connected with application of diffusion maps to dimensionality reduction and data visualization as described in [10,11,12].

The first step of PHATE algorithm, for a d-dimensional data space $X=\left\{x_{1},x_{2},\ldots ,\;x_{N}\right\}$ is calculating similarities between points using Gaussian kernel:(5)$$k_{\varepsilon }\left(x,y\right)=\exp\left(-\frac{∥x-y{∥}^{2}}{\varepsilon }\right)$$
ε is the radius that depends on the spread of neighbors captured by the kernel. The kernel is normalized
(6)$$v_{\varepsilon }\left(x\right)=∥k_{\varepsilon }{(x,\;\cdot )∥}_{1}=\sum_{j=1}^{N}k_{\varepsilon }\left(x,x_{j}\right)$$

And we get an N × N stochastic matrix with entries
(7)$${\left[P_{\varepsilon }\right]}_{\left(x,y\right)}=\frac{k_{\varepsilon }\left(x,y\right)}{v_{\varepsilon }\left(x\right)}$$

That are probabilities of moving from x to y in one step. Afterwards, the radius $\varepsilon_{k}\left(x\right)$ (bandwidth) is adjusted using for each data point x the k-NN distance. This results in α decaying kernel
(8)$$K_{k,\alpha }\left(x,y\right)=\frac{1}{2}\exp\left(-{\left(\frac{∥x-y{∥}_{2}}{\varepsilon_{k}\left(x\right)}\right)}^{\alpha }\right)+\frac{1}{2}\exp\left(-{\left(\frac{∥x-y{∥}_{2}}{\varepsilon_{k}\left(y\right)}\right)}^{\alpha }\right),$$

α in an exponent that controls the rate of decay of the tails in kernel.

The normalized kernels form a probability transition matrix *P*, a diffusion operator that defines Markovian ergodic diffusion process. We consider diffusion distance given by
(9)$$D^{t}\left(x,y\right)={\left(\overset{N}{\underset{j=1}{\sum }}\frac{{\left(p_{x}^{t}\left(x_{j}\right)-p_{y}^{t}\left(y_{j}\right)\right)}^{2}}{p_{x}^{\infty }\left(x_{j}\right)}\right)}^{1/2},$$

The diffusion time scale *t* is determined using von Neuman entropy. Parameter *t* determines the number of steps taken in a random walk and is a trade-off between encoding local and global information in the embedding.

Afterwards, we define t-step potential distance between point x and y. Namely, the Markov process that defines the diffusion geometry converges asymptotically to a diffusion process governed by Fokker–Planck equations with a potential 2U(x) [10]. Using random walk with t-steps and a fixed initial condition as a data generation process, we obtain that the generated data is distributed according to $p_{x}^{t}$ and the corresponding t-step potential representation of x is $U_{\varepsilon ,x}^{t}=-\log\left(p_{x}^{t}\right)$. The distance is given by
(10)$$B^{t}\left(x,y\right)=∥U_{x}^{t}-U_{y}^{t}∥{}_{2}$$

The distances are then used in the multi-dimensional scaling (MDS), the final step of the algorithm for dimensional reduction needed in visualization procedure.

In our research, we first performed DEA to obtain homogeneous groups of objects. Then, we applied archetypal analysis to find extreme observations in the set and references for other objects in relevant DEA groups. Archetypal analysis provided a new interpretation for DEA results. Finally, we visualized the results with help of PHATE in a 2- and 3-dimensional space.

## 3. Data

We compared the described methods on a set of 68 production companies trading on the Warsaw Stock Exchange (WSE). We used averages of quarterly values for financial indicators for 2011 and 2012, published by Notoria Service, to describe our objects. In analysis 1, we used six financial indicators: asset turnover and total liabilities/total assets (debt ratio) as input indicators (indicators for which values are preferred) and return on assets (ROA), return on equity (ROE), current ratio (CR), operating profit margin (OPM) as output indicators (high values are preferred) for the DEA model. The choice was determined by the strict requirements of DEA models and our previous experience [9]. In analysis 2, we used 21 financial indicators to describe the companies under consideration.

## 4. Results

First, we performed DEA for the set of 68 companies described with six financial indicators. In order to obtain division into homogeneous groups of companies, we performed the DEA algorithm on the whole set of DMUs. The efficient units with efficiency score 1 constitute the first homogeneous group. After removing all efficient units, we applied the DEA algorithm again, distinguishing in this way the next group of efficient units. The procedure was repeated until six groups of objects were found. The first group consisted of the best 10 companies. For these companies ROA, ROE, CR and OPM values were high and DR and AT low. Next, we found archetypes for this set of companies using unitarized values of indicators. To visualize the observations and detect similarities between them, we used the PHATE algorithm. Archetypes were visualized as objects in this space. Apart from archetypes, the DEA groups were visualized in this space. The same procedure of archetypal analysis and PHATE was repeated for the set of 68 companies described this time with 21 financial indicators.

The calculations were done in SAS (ver. 9. 4) Python (ver. 3.6) and R (ver. 4.0.5).

### 4.1. Analysis 1

First, we performed DEA for the set of 68 companies described with six financial indicators. We distinguished six groups of homogeneous objects. The first group consisted of the best 10 companies. For these companies ROA, ROE, CR and OPM values were high and DR and AT low. Next, we found archetypes for this set of companies using unitarized values of indicators. We distinguished three archetypes. In Table 1, the percentile values of each archetype are presented as percentiles of maximal values of financial indicators.

Figure 1 displays bar plots (i.e., the percentile value of each indicator for each archetype). Archetype 1 represents a company with quite a high value of AT, very low values of DR, ROA, ROE and OPM and a moderate value of CR. There are two companies that represent this archetype: MOJ and TAURON. The percentiles of maximal values for these companies are given in Table 2. Both firms belong to DEA group 4.

Archetype 2 represents a firm with a low value of AT but quite a high value of DR, as shown in Table 3. The other values are moderate ranging from 20% to 66%. This archetype is represented by RAFAKO and SYNEKTIK, both in DEA group 4.

Archetype 3 can be easily interpreted. This archetype has low values of AT and DR and high values of other indicators as shown in Table 4. An existing company that exactly matched the archetype could not be found, but there are two companies very close to it: AC with weight 0.95 and EKO EXP with weight 0.94, both belong to DEA group 1.

All companies that are efficient or have high efficiency score but IZOL_JAR and KPPD are close to archetype 3. Objects in DEA group 2 are close to archetype 3. On the other hand, the companies that are not efficient (groups 4, 5 and 6) are linear combinations of archetype 1 and archetype 2.

The next step was to perform the PHATE algorithm to visualize the objects in a 2-dimensional space and in a 3-dimensional space. Figure 2 and Figure 3 show the results. PHATE confirmed the relations that were discovered by archetype analysis and properly captured the connections to DEA groups 1 and 2. Archetypes are not only extreme observations but representatives of certain groups of objects. PHATE provides good insight into data structure as it shows clusters of objects that are related or close with respect to DEA.

### 4.2. Analysis 2

In Analysis 2, that was performed on the same set of 68 companies described by 21 financial indicators, we distinguished 3 archetypes, as shown in Figure 4.

Archetype 1 can be easily interpreted. This archetype represents a firm with low values of DR, EBIT, RT and RC, moderate values of AT and OC and high values of other indicators. There are two companies that exactly match the archetype, AC and EKO_EXP with weight 1, both in DEA group 1. Efficient objects (DEA groups 1 and 2) are close to archetype 1.

Archetype 2 represents a firm that has high values of DR, EBIT, WC, moderate values of QR2 and RC and low values of the other indicators. There are four companies that exactly match the archetype DUDA, KPPD, LOTOS and MIESZKO with weights 1 and ERG with weight 0.99. There is one company that exactly matches Archetype 3. It is MUZA, with weight 1. The company belongs to DEA group 4. Archetype 3 has low values of ROA, ROE, OPM and DSR, moderate values of DR, QR2 and EBIT and high values of other indicators. Inefficient objects (DEA groups 4, 5 and 6) are linear combinations of archetypes 2 and 3.

The next step was to perform the PHATE algorithm to visualize the objects in 2- and 3-dimensional spaces. Figure 5 and Figure 6 show the results. Again, PHATE confirms the relations that were discovered by archetype analysis and properly captures the connections to DEA. Archetypes are again not only extreme observations but representatives of certain groups of objects. Efficient objects (DEA group 1 and 2) are close to archetype 1.

## 5. Discussion

The aim of our research was to combine the DEA method and archetypal analysis treated as segmentation methods with the PHATE visualization algorithm.

Archetypal analysis is a tool in multivariate data analysis used directly for segmentation. In many fields of research, segmentation is a very useful tool as the first step to typology. The analysis of archetypes allows us to look at the typology from a different point of view. Optimal archetypes define a convex data region so that observed objects can be expressed in terms of a set of weights. The weights make it possible to assess the similarity to individual archetypes, which thus acquire the character of benchmarks. This work uses this technique to classify enterprises.

In this paper, the archetypes were compared with the results obtained by the DEA BCC model. Due to the different mathematical models underlying both methods, a comparison using the statistical methodology does not seem possible or proper.

For the purpose of comparison, visualization techniques based on the form of a low-dimensional maps were applied. Classical methods such as PCA (principal components analysis) or MDS (multidimensional scaling) proved to be of little use due to the fact that they use the global properties of units. The PHATE method allowed the local properties of units through non-linear data transformations based on the probability distribution of economic indicators to be taken into account, and then employed the projection to a smaller dimension. The final result showed the objects in space with the preservation of their similarities defined as distances to the closest neighbors on the maps.

The analyses were carried out for two sets of KPIs (key performance indexes). The first one included only the most important indicators from the economic point of view, while the second one broadened the scope of indicators to those often used in practice.

The figures contained in the paper show the segments identified by DEA and the archetypes obtained from the archetypal analysis. It is worth noting that the DEA technique and the classification of enterprises based on archetypal analysis gave consistent results. As a consequence, the method allows for the transformation of archetypes covering the observed space into a space of economic efficiency in terms of trend makers, as well as defining the desired patterns and introducing a metric in this space.

Associating DEA segmentation with archetypes sheds new light on the interpretation of the results. The proposed approach can be used to get a better insight into the data structure with archetypes—the potential trend makers—taking into account the economic efficiency of objects.

To our knowledge, there is no similar analysis that would combine other methods with PHATE visualization. Application of DEA and archetypal analysis enriched the PHATE results and made them easier to interpret.

The proposed approach could find widespread applications in various fields dealing with multidimensional objects.

## Figures and Tables

**Figure 1 entropy-24-00088-f001:**
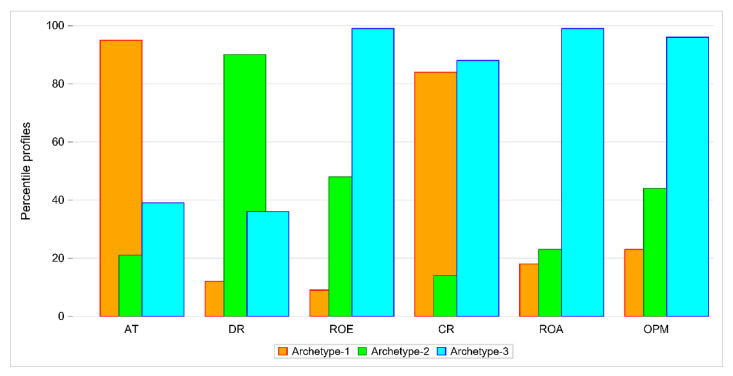
Comparison of three archetypes with respect to percentile values of indicators.

**Figure 2 entropy-24-00088-f002:**
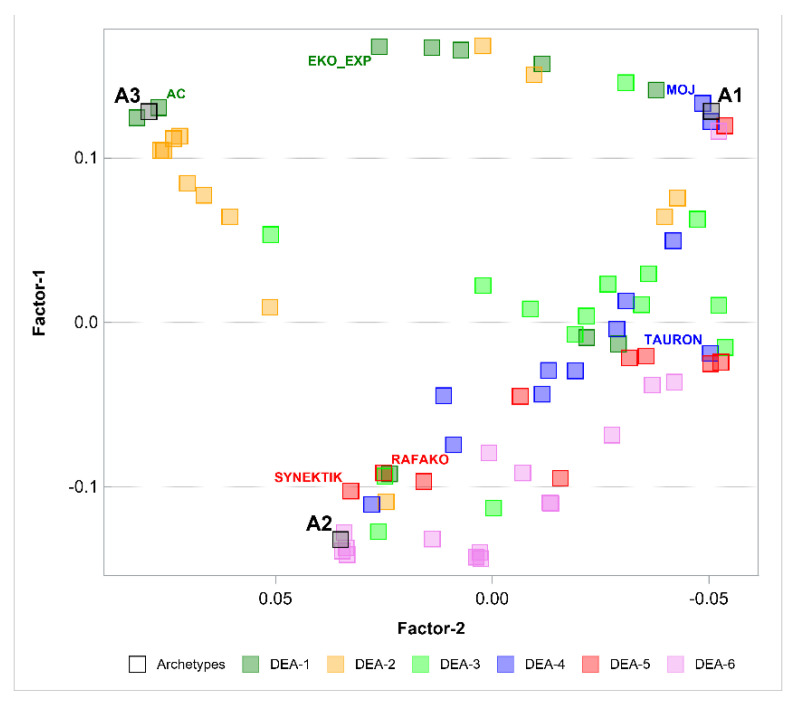
Archetypes in 2-dimensional space generated by PHATE to visualize objects. Archetypes and DEA groups are shown.

**Figure 3 entropy-24-00088-f003:**
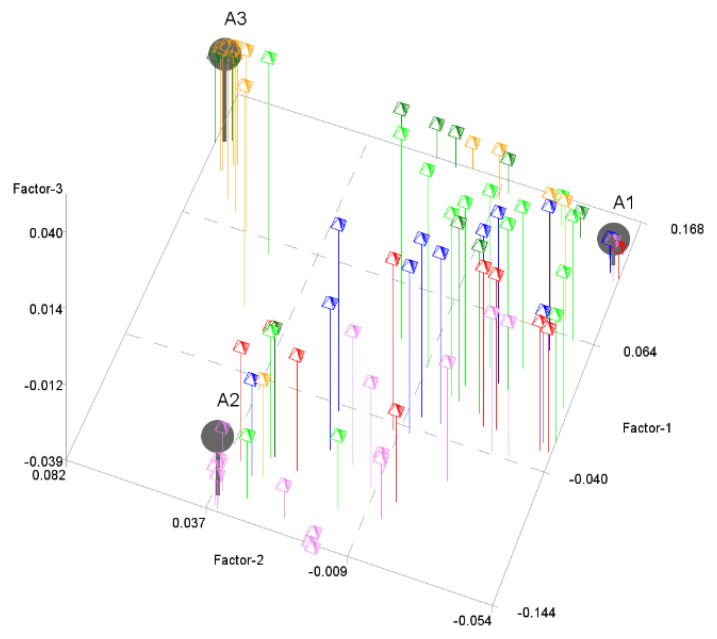
Archetypes in 3-dimensional space generated by PHATE to visualize objects. Archetypes and DEA groups are shown.

**Figure 4 entropy-24-00088-f004:**
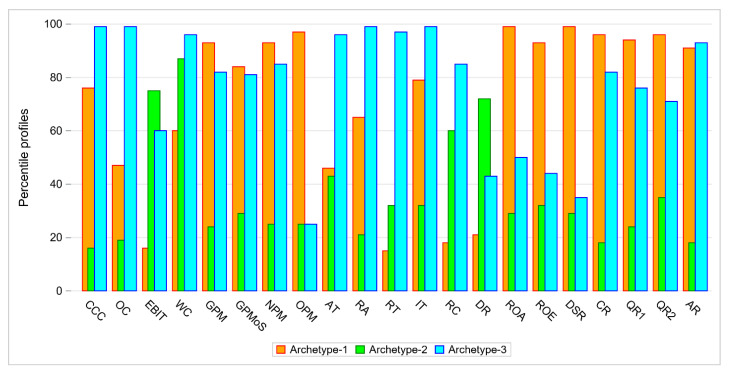
Comparison of archetypes with respect to percentile values of 21 indicators.

**Figure 5 entropy-24-00088-f005:**
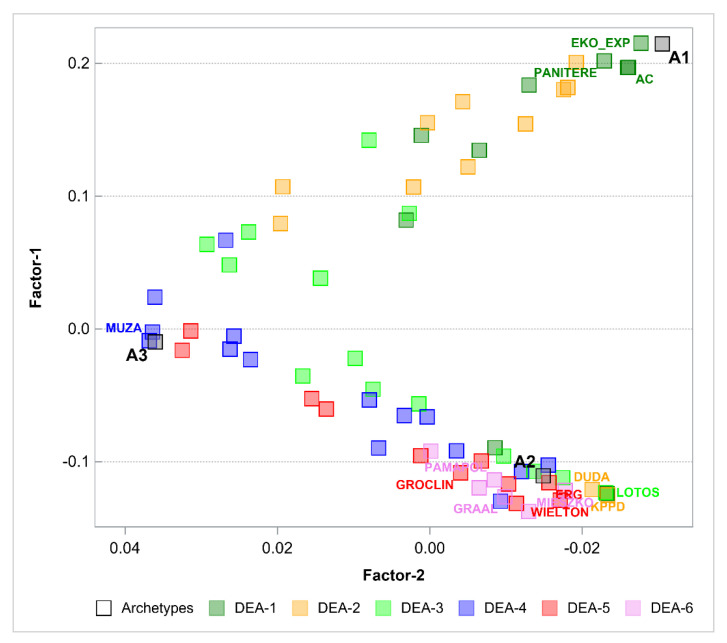
Archetypes in 2-dimensional space generated by PHATE to visualize objects. Archetypes and DEA groups are shown.

**Figure 6 entropy-24-00088-f006:**
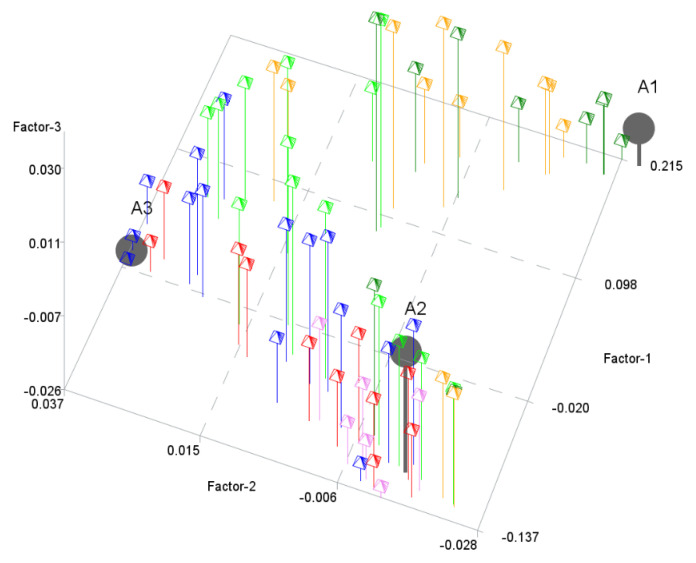
Archetypes in 3-dimensional space generated by PHATE to visualize objects. Archetypes and DEA groups are shown.

**Table 1 entropy-24-00088-t001:** The percentile value in an archetype as compared to the maximum value of the variable.

Archetype	AT	DR	ROA	ROE	CR	OPM
Archetype 1	91	21	10	9	72	13
Archetype 2	18	91	40	66	21	47
Archetype 3	43	25	99	97	96	97

**Table 2 entropy-24-00088-t002:** The fractions of maximal values of each indicator for Archetype 1 representatives.

	AT	DR	ROA	ROE	CR	OPM
MOJ	0.67	0.32	0.07	0.06	0.47	0.04
TAURON	0.77	0.32	0.20	0.17	0.15	0.02

**Table 3 entropy-24-00088-t003:** The fractions of maximal values of each indicator for Archetype 2 representatives.

	AT	DR	ROA	ROE	CR	OPM
RAFAKO	0.32	0.96	0.17	0.32	0.15	0.30
SYNEKTIK	0.27	0.98	0.16	0.33	0.17	0.24

**Table 4 entropy-24-00088-t004:** The fractions of maximal values of each indicator for Archetype 3 representatives.

	AT	DR	ROA	ROE	CR	OPM
AC	0.23	0.43	1	1	0.52	0.71
EKO EXP	0.45	0.20	0.7	0.55	1	1

## Data Availability

Financial reports were sources from https://ir.notoria.pl/ (accessed on 23 November 2014).

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
