# Peer review of "Archetypal Analysis and DEA Model, Their Application on Financial Data and Visualization with PHATE"

_entropy, 2022, doi:10.3390/e24010088_

Round 1

Reviewer 1 Report

  1. Results: Recommend to be Rejected

This paper compares the results of the typology of enterprises based on both methods. The DEA BCC model and Archetypal Analysis are applied to companies listed on the Warsaw Stock Exchange and described by financial indicators (KPIs). The authors show that both methods give consistent results and propose how to use Archetypal Analysis to create patterns of effectiveness. The authors used a new approach called PHATE algorithm to reduce the dimensions and visualize the results of DEA and Archetypal Analysis.

This paper is with minor merits for Entropy, due to the old methods proposed in previous paper, however, it requires some major revisions.

Firstly, the abstract should be refined to clearly indicate what authors had done within 150 words. The abstract is saying nothing (lots of garbage wordings) to keep what authors had really done.

Secondly, for Section 1, authors should provide the comments of the cited papers after introducing each relevant work. What readers require is, by convinced literature review, to understand the clear thinking/consideration why the proposed approach can reach more convinced results. This is the very contribution from authors. In addition, authors also should provide more sufficient critical literature review to indicate the drawbacks of existed approaches, then, well define the main stream of research direction, how did those previous studies perform? Employ which methodologies? Which problem still requires to be solved? Why is the proposed approach suitable to be used to solve the critical problem? We need more convinced literature reviews to indicate clearly the state-of-the-art development.

For Section 2, please condense them into one or two relevant sections. In addition, authors should introduce their proposed research framework more effective, i.e., some essential brief explanation vis-à-vis the text with a total research flowchart or framework diagram for each proposed algorithm to indicate how these employed models are working to receive the experimental results. It is difficult to understand how the proposed approaches are working.

For Sections 3 and 4, authors should use more alternative models as the benchmarking models, authors should also conduct some statistical test to ensure the superiority of the proposed approach, i.e., how could authors ensure that their results are superior to others? Meanwhile, authors also have to provide some insight discussion of the results. Authors can refer the following reference for conducting statistical test.

Forecasting short-term electricity load using hybrid support vector regression with grey catastrophe and random forest modeling. Utilities Policy, 2021, 73, 101294.

Author Response

Dear Reviewer,

We should first of all thank you very much for reading our manuscript and for all valuable remarks and comments.

We have introduced changes in the manuscript. The title, abstract and discussion have been changed according to your suggestions. Some correction in the text have been made. We hope that we have managed to stresses sufficiently our research contribution and to describe better our research design.

We have added only two new papers. It is difficult to add any new papers about DEA as there are hundreds of papers published annually on it.

We hope that the amendments improve the quality of our paper.

With best regards,

Urszula Grzybowska and Marek Karwanski

Reviewer 2 Report

The paper entitled “Archetypal Analysis and classical segmentation methods. Comparison of two approaches on financial data by  Urszula Grzybowska et al. compares the results of the typology of enterprises based on both methods (archetypal analysis and DEA).  Also the authors proposed a non-linear PHATE method to reduce the dimensions and visualize the results  of DEA and Archetypal Analysis. The paper contains new results by evaluating data from Warsaw Stock Market,  using these methods. The paper could be published taking into account the following remarks:

  1. It is not clear the contribution of this paper to the economic analysis. More specifically if the new contribution is the PHATE model the authors have to show exactly their contribution to this methods.
  2. The reference should be updated. More specifically, only three out of ten references are during last decade.
  3. The authors have to show more clearly the agreement among the models.
  4. It is necessary to be explained which is the meaning of results in figures. What does echa factor reveal?

Author Response

(The authors gave the same response as above.)

Reviewer 3 Report

The paper uses Data Envelopment Analysis to identify drivers and barriers of group firms grouped based on their economic efficiency. The paper is of potential interest for Entropy readers, although I found many major issues which prevent me to accept the paper in the current form:

  1. Authors have to provide more emphasis on the novelty of their work compared to existing ones. It is not clear enough the genuine contribution of the current work compared to extant studies in the literature. Thus, the introduction section needs to be completely rewritten and expanded by using recent references.
  2. Authors may offer to the reader additional information about their decision to explore use 68 production companies traded on Warsaw Stock Exchange. Besides that, data are outdated (2011-2012) lowering the relevance of authors’ findings which may not apply 10 years later.
  3. The discussion section is largely underdeveloped. Authors, authors have to provide more discussion comparing and contrasting their results from existing studies pointing out the novelty of their work. This section needs to be deeply expanded.
  4. A large part of the conclusions is not appropriate, as it is only a concise repetition of the comments. In the conclusions, you should simultaneously consider all you have discovered, and exploit it to add something new (or new interpretations), and policy indications.
  5. The authors do not discuss possible limitations of their study or the insights for future directions of research. Maybe they could discuss external validity of the results in terms of possible insights in other countries and/or additional variables that they would have liked to have to better answer to their research question.
  6. I recommend that authors review the article thoroughly and consider using a professional proofreading service to improve the style of the article. Many sentences are unclear.
  7. I suggest the authors to rewrite the introduction and the discussion section using recent references.

Author Response

(The authors gave the same response as above.)

Round 2

Reviewer 1 Report

Authors have completely addressed all my concerns.

Reviewer 2 Report

The paper entitled “Archetypal Analysis and classical segmentation methods. Comparison of two approaches on financial data by  Urszula Grzybowska et al. now can be published since it has adopted the main part of previous review.

Reviewer 3 Report

The manuscript has improved compared to the previous version and can be accepted for publication in the current form.